# Unveiling Lovastatin’s Anti-Inflammatory Potential in Mouse’s Brain during Acute *Trypanosoma cruzi* Infection

**DOI:** 10.3390/biology13050301

**Published:** 2024-04-27

**Authors:** Beatriz Matheus de Souza Gonzaga, Líndice Mitie Nisimura, Laura Lacerda Coelho, Roberto Rodrigues Ferreira, Samuel Iwao Maia Horita, Daniela Gois Beghini, Vanessa Estato, Tania Cremonini de Araújo-Jorge, Luciana Ribeiro Garzoni

**Affiliations:** 1Laboratório de Inovações Em Terapias, Ensino E Bioprodutos, Instituto Oswaldo Cruz, Fundação Oswaldo Cruz, Rio de Janeiro 21040-900, RJ, Brazil; biagonzaga04@hotmail.com (B.M.d.S.G.); lindicem@gmail.com (L.M.N.); llacerdac@gmail.com (L.L.C.); robertoferreira@ioc.fiocruz.br (R.R.F.); samuelhorita@gmail.com (S.I.M.H.); beghini@ioc.fiocruz.br (D.G.B.); taniaaj@ioc.fiocruz.br (T.C.d.A.-J.); 2Laboratório de Genômica Aplicada e Bioinovações, Instituto Oswaldo Cruz, Fundação Oswaldo Cruz, Rio de Janeiro 21040-900, RJ, Brazil; 3Laboratório de Pesquisa do Timo, Instituto Oswaldo Cruz, Fundação Oswaldo Cruz, Rio de Janeiro 21040-900, RJ, Brazil; 4Laboratório de Imunofarmacologia, Instituto Oswaldo Cruz, Fundação Oswaldo Cruz, Rio de Janeiro 21040-361, RJ, Brazil; vanessaestato@gmail.com

**Keywords:** chagas disease, brain microcirculation, lovastatin, anti-inflammatory activity

## Abstract

**Simple Summary:**

Cerebral alterations have been identified in Chagas disease since its discovery. Meningoencephalitis and stroke were verified in patients and experimental models of infection. Our group previously described brain microvasculopathy in mice caused by *Trypanosoma cruzi* (Y strain) infection, which was characterized by endothelial dysfunction, a reduction in perfused capillaries, and increased leukocyte rolling and adhesion. Nowadays, nifurtimox and benznidazole are still the only options for Chagas disease treatment, but the high rate of side-effect occurrence may lead patients to treatment interruption. In the search for better treatment alternatives, statins show promising effects. The literature shows evidence that statins present anti-inflammatory activity and the ability to improve endothelial function, and they could also have trypanocidal activity. All these indicate that it could be an interesting drug to treat cerebral microvasculopathy in Chagas disease. In this paper, we investigate the effect of lovastatin on microcirculation damage and brain inflammation caused by acute experimental *T. cruzi* infection. Here, we report that lovastatin prevented the increase in F4/80+ cells and ICAM-1 levels in the brain caused by acute infection with *T. cruzi*, suggesting an anti-inflammatory activity of lovastatin.

**Abstract:**

Neurological commitment is a neglected manifestation of Chagas disease (CD). Meningoencephalitis mainly affects children and immunosuppressed patients, while stroke can occur with or without cardiac compromise. One of the possible causes of stroke development is microvascular commitment. Our group previously described that experimental *Trypanossoma cruzi* acute infection leads to cerebral microvasculopathy. This condition is characterized by decreased capillary density, increased leukocyte rolling and adhesion, and endothelial dysfunction. CD was discovered 114 years ago, and until today, only two drugs have been available for clinical treatment: benznidazole and nifurtimox. Both present a high cure rate for the acute phase (80%) and small cure rate for the chronic phase (20%). In addition, the high occurrence of side-effects, without proper medical follow-up, can result in treatment abandonment. Therefore, the search for new therapeutic schemes is necessary. Statins are drugs already used in the clinic that have several pleiotropic effects including endothelial function improvement, anti-inflammatory action, as well as trypanocidal effects, making them a potential alternative treatment for brain microvasculopathy in CD. Here, we investigate the effect of lovastatin (LOV) on brain microvasculopathy and inflammatory parameters. Swiss Webster mice were intraperitoneally inoculated with the Y strain of *T. cruzi*. Treatment with lovastatin (20 mg/kg/day) was initiated 24 h after the infection and continued for 14 consecutive days. We observed that LOV treatment did not affect parasitemia, brain microcirculation alterations, or the reduction in cerebral blood flow caused by *T. cruzi* infection. Also, LOV did not prevent the increased number of CD3+ cells and eNOS levels in the *T. cruzi*-infected brain. No alterations were observed on VCAM-1 and MCP-1 expressions, neither caused by infection nor LOV treatment. However, LOV prevented the increase in F4/80+ cells and ICAM-1 levels in the brain caused by acute infection with *T. cruzi*. These results suggest an anti-inflammatory activity of LOV, but more studies are needed to elucidate the role of LOV in CD acute infection.

## 1. Introduction

Chagas disease (CD) is a protozoan infection caused by *Trypanosoma cruzi.* It affects around 6 million people around the globe and 75 million people are still at risk of infection [1,2]. CD has two phases: acute and chronic. The acute phase is commonly asymptomatic or presents non-specific symptoms, which makes the diagnosis in this phase more difficult [3]. The main cause of death is myocarditis and meningoencephalitis [4,5]. In the chronic phase, patients could remain in the indeterminate form of the disease for many years. The main form of CD in this phase is chronic Chagas cardiomyopathy, but digestive manifestations could also occur [6]. Also, neurological symptoms have been described since the discovery of CD [7,8]. One of the signs of cerebral damage in the chronic phase is the occurrence of stroke in patients with or without cardiac commitment [9].

Meningoencephalitis is a severe condition characterized by the presence of inflammatory cells in the brain and meninges [10]. In CD, it occurs mainly in children and immunosuppressed patients in the exacerbation of the disease [11]. Histological analysis of human brain sections showed encephalitis with the presence of inflammatory nodules composed of mononuclear cells, glial cells, and *T. cruzi* nests [12]. Also, in an experimental model of CD, it was demonstrated that C3H/HE mice infected with the Colombian strain presents meningoencephalitis restricted to the acute phase of the disease, with lymphocyte infiltration, composed mainly by CD8+ T cells, not co-localized with *T. cruzi* antigens [13].

Stroke is another condition associated with CD that deserves more attention [9]. According to a recent study, approximately 40% of CD patients are diagnosed with the disease after suffering their first stroke [14]. Moreover, studies indicate that 25% of stroke cases in CD are not related to cardioembolism [15]. However, several non-embolic causes could be related to the occurrence of stroke, such as cryptogenic (25.5%), microcirculatory disease (9.5%), and large vessel atherosclerosis (8.5%) [15]. Additionally, pathological studies performed on the brains of patients with advanced chronic disease showed that in approximately 25% of cases, the presence of cerebral infarct areas could be observed [16].

Our group demonstrated for the first time the existence of cerebral microvasculopathy in a murine model of acute *T. cruzi* infection. This condition was characterized by a decreased number of spontaneously perfused capillaries and the increased rolling and adhesion of leukocytes in the venules, in addition to endothelial dysfunction and increased aggregation platelets with microvascular thrombus formation [17]. A more recent study from our group demonstrated in the same animal model that *T. cruzi* infection also causes a reduction in brain blood flow and increases the number of F4/80+ and CD3+ cells in brain tissue. We also verified that treatment with benznidazole in either the usual dose or reduced dose prevented the alterations caused by the infection to microcirculation and inflammatory parameters [18].

Benznidazole is the drug of choice to treat CD [19], but collateral effects could be observed in almost 40% of the patients during treatment [20]. In these patients, the lack of proper medical care could lead to the interruption of treatment without completing the therapeutic scheme [21]. Statins are drugs clinically used for the treatment of hypercholesterolemia with pleiotropic effects described by the literature [22]. Simvastatin, lovastatin, pitavastatin, and atorvastatin are lipophilic statins, which indicates that they are more likely to cross the blood–brain barrier, suggesting a better activity on conditions that affect the brain [23]. Studies have demonstrated that statins present trypanocidal activity both in vitro [24] and in vivo using experimental models of acute CD [25,26]. Also, statins have antioxidant and immunomodulatory effects, can improve endothelial function, inhibit the inflammatory response, increase endothelial levels of nitric oxide (NO), and improve atherosclerotic plaque stability [27]. However, there is no study elucidating the effect of lovastatin on microvascular damage caused by *T. cruzi* infection.

In this paper, we investigate the effect of lovastatin treatment on microvasculopathy, inflammation, and inflammatory mediators in the brain of mice during acute *T. cruzi* infection. We hope to bring more information on the use of lovastatin as a potential alternative treatment for brain microvasculopathy in CD.

## 2. Materials and Methods

### 2.1. Ethics Statement

Animal utilization and experimental protocols are in compliance with Brazilian Law 11.794/2008 and with the National Council of Animal Experimentation Control (CONCEA). All procedures received approval from the Oswaldo Cruz Foundation Animal Welfare Committee (License number LW-21/16) and are in accordance with the guideline of the USA National Institutes of Health Guide for the Care and Use of Laboratory Animals (NIH Publication No. 85-23).

### 2.2. Animals and Parasite

For this study, Swiss Webster male mice aged 6 to 8 weeks old were used, with weights of 18 to 20 g, obtained from the Animal Facilities of Oswaldo Cruz Foundation (ICTB, Rio de Janeiro, Brazil). Animals were maintained in controlled conditions (12/12 h light–dark cycle/temperature of 22 °C), for a week before any procedure. Mice were intraperitonially inoculated with 5 × 10^4^ trypomastigotes of *T. cruzi* (Y strain—TcII DTU).

### 2.3. Experimental Groups

Mice were randomly divided into four groups. For each infected group, we had a matched non-infected group, housed in identical conditions. For the non-infected groups, 3–4 animals were allocated per group, while for groups of infected animals, the number was 6–8 animals per group. The experimental groups used in the evaluations were non-infected non-treated (NINT), non-infected treated with lovastatin (NI + LOV), non-treated infected with *T. cruzi* (YNT), and infected treated with lovastatin (Y + LOV). To assess parasitemia and weight, three independent experiments were conducted. For assessment of capillary density, leukocyte interactions, and brain blood flow, three experiments were performed. For immunohistochemistry and Western blot analysis, two experiments were performed.

### 2.4. Drug and Treatment Scheme

Lovastatin (SANDOZ, Sao Paulo, Brazil) was diluted in distilled water and administered at the usual dose of 20 mg/kg. Medication was orally administered in the morning by gavage. An early treatment protocol was used as previously described [18], initiated 1 day post infection and maintained for 14 days.

### 2.5. Parasitemia, Body Weight, and Survival

For parasitemia, we used the Pizzi–Brener method which consists of counting parasites in 5 µL of tail blood of each mouse by direct microscopic examination. It was performed from 6 to 10 days post infection to follow-up the peak and decay of blood parasites. Body weight was monitored three days a week for fifteen days post infection. Survival was regularly monitored for twenty-two days post infection (dpi) in three independent experiments of 4 animals each.

### 2.6. Intravital Video Microscopy

The animals used for parasitemia and body weight analysis were anesthetized at 15 dpi with a mixture of xylazine (10 mg/kg) and ketamine hydrochloride (75 mg/kg), intraperitonially. We used a stereotaxic frame to immobilize the mice and perform the surgery. A craniotomy was performed using a high-speed drill to expose a cranial window. Then, animals were placed on an intravital microscope with a mercury lamp (Olympus BX51/WI, Sanford, NC, USA) attached to a CCD digital video camera system to assess brain microcirculation. The evaluation of the images obtained was blind, using a code number as previously described [18].

### 2.7. Assessment of Functional Capillary Density

Intravenous injection of the fluorescent dye FITC-labeled dextran (5%) (0.1 mL) was administered. Images were obtained by Archimed 3.7.0 software (Microvision, Evry, France), which allowed us to visualize the capillaries in real-time online using Saisam 5.1.3 software (Microvision, Evry, France). The total number of spontaneously perfused capillaries (functional capillary density) was counted and represented vessels with diameters less than 10 mm per square mm of surface area (1 mm^2^).

### 2.8. Leukocyte Rolling and Adhesion Analysis

The intravenous injection of rhodamine 6G (0.3 mg/kg) permitted the visualization of leukocytes. We evaluated a segment of 100 mm length of three randomly selected venulas with 30 to 100 mm diameters for 60 s each. For this analysis, we considered rolling leukocyte cells that crossed the venula slower than circulating red blood cells, and adherent leukocytes were considered cells that did not move for 30 s, attached to the venular wall.

### 2.9. Analysis of Brain Blood Flow

We used a laser speckle contrast imaging system (LSCI) (Perimed, Jarfalla, Sweden) to evaluate the mice’s brain blood flow. Animals were anesthetized with ketamine and xylazine to perform craniotomy. Following that, animals were placed under an LSCI light (wavelength of 785 nm), maintaining a distance of 10 cm between the laser light and the mice’s brain. Brain blood flow was assessed in real-time. After defining a region of interest (ROI), six laser speckle images were obtained per second and the relative brain blood flow was assessed using Perisof 5001 software (Perimed, Jarfalla, Sweden) and is expressed as arbitrary perfusion units (APUs). This evaluation was blind, as described previously [18].

### 2.10. Immunohistochemistry

Tissues were obtained from some of the animals previously used for intravital microscopy. Cryosections of 5 μm of the brain encephalic region were obtained and fixed with ice-cold acetone (Merck, Rahway, NJ, USA) for 10 min. The blockage of non-specific binding was made using a goat serum (Vector Laboratories, Newark, NJ, USA) for 20 min. F4/80 clone BM8 and CD3 clone 17A2 (Biolegend, San Diego, CA, USA) were used in the analysis and were incubated overnight after dilution on PBST. The ImmPRESS^®^ HRP Goat Anti-Rat IgG Polymer Detection Kit with DAB (Dako, Santa Clara, CA, USA) was used to detect the antibodies. The counterstain used was Harris’s Hematoxylin Solution (EasyPath-São Paulo, Brazil) for 1 min. The slides were rinsed with tap water and then dehydrated chemically and mounted with Erv-mount mounting media (EasyPath). The counting of CD3+ and F4/80+ cells was performed manually. Representative images were acquired in Motic Panthera L Microscope Built-in Smart CAM & ImagingOnDevice System (Motic-Xiamen, Xiamen, China).

### 2.11. Western Blot

Tissues were obtained from some of the animals previously used for intravital microscopy. An amount of 50 mg of brain tissue (without cerebellum and olfactory bulb) was weighed and added to 500 µL of lysis buffer [25 mM Tris HCl, 150 mM NaCl, 1 mM EDTA, 50 mM NaF, 1 mM Na_3_VO_4_, 1% Triton].

Protein measurement was performed using spectrophotometry using the BCA kit (Thermo Scientific Pierce, Waltham, MA, USA), according to the manufacturer’s instructions. The absorbance of the total proteins in the samples was compared to the BSA standard curve. The extracts were maintained at −80 °C.

An amount of 20 or 100 µg of proteins was diluted in a sample buffer and subjected to a temperature of 100 °C for 5 min. A polyacrylamide gel containing sodium dodecyl sulfate (SDS-PAGE) was used. An electric field of 120 volts was applied. Samples were transferred to a nitrocellulose membrane using a semi-dry transference system (Bio-rad, Hercules, CA, USA). To inhibit the binding of antibodies to nonspecific proteins, we used a blockage solution containing 5% of milk for 1 h. Primary antibodies (Table 1) were incubated overnight at 4 °C, with the exception of the antibody anti-GAPDH that was incubated for 1 h at room temperature, as well as the secondary antibodies.

All antibodies were diluted in a blocking solution. Peroxidase development was performed by chemiluminescence, using the kit SuperSignal West Pico (Thermo Scientific) and X-ray films. The densitometry was carried out with the program Image Studio Lite Ver 4.0.

### 2.12. Statistical Analysis

Unpaired *t*-tests or analysis of variance (ANOVA) followed by Tukey’s multiple comparison test were used for data analysis. Results are shown as mean ± SEM for each group. We considered statistically significant results that presented differences with *p* values of less than 0.05. The GraphPad Prism statistical package (GraphPad InStat 5.0, GraphPad Software Inc., La Jolla, CA, USA) was used for all calculations.

## 3. Results

### 3.1. Effect of Lovastatin on Parasitemia, Body Weight, and Survival during T. cruzi Infection in Mice

Swiss Webster mice were inoculated with 10^4^ trypomastigote forms of the Y strain of *T. cruzi* intraperitoneally and treated with 20 mg/kg/day of lovastatin. They were divided into four groups: non-infected non-treated mice (NINT), non-infected treated mice (NI + LOV), infected non-treated mice (YNT), and infected treated mice. The first step of our analysis was to investigate the effect of LOV on experimental characterization. Evaluating parasitemia, we observed that the YNT group showed a high peak of parasitemia at 8 dpi with a mean of 2.145 × 10^7^ parasites/mL (Figure 1A). The Y + LOV group, treated with 20 mg/kg/day lovastatin, also presented a parasitemia peak at 8 dpi with a mean of 2.628 × 10^7^ parasites/mL and did not present a significant difference when compared to the infected untreated group (YNT).

Analysis of the mice’s body weight (Figure 1B) showed no difference in this parameter. At 15 dpi, the NINT control group presented a body weight of 37.89 ± 1.28 g, the NI + LOV group 37.65 ± 0.97, the YNT group 30.10 ± 1.08 g, and the Y + LOV group 29.40 ± 0.82 g.

Treatment with LOV did not influence the high mortality caused by the infection (Figure 1C). At 22 dpi only two mice lived out of twelve in both YNT and Y + LOV groups.

### 3.2. Influence of Lovastatin on Cerebral Microvasculopathy Caused by T. cruzi Infection in Mice

We previously showed that acute *T. cruzi* infection causes cerebral microvasculopathy in mice [17] and treatment with BZ prevented this alteration [18]. Here, we evaluate the effect of lovastatin treatment on capillary density. We observed that the YNT group (471.25 ± 73.18 capillaries/mm^2^) (Figure 2C) presented decreased microvascular perfusion when compared with NINT (672 ± 78.08 capillaries/mm2, *p* < 0.001) (Figure 2A). At 15 dpi, the Y + LOV group presented 543.91 ± 72.24 capillaries/mm2 and was not distinct from the YNT group. The capillary density of the NI + LOV group did not change in comparison to the NINT control group (Figure 2B).

### 3.3. Effect of Lovastatin Treatment on Cerebral Leukocyte–Endothelium Interactions during Acute T. cruzi Infection in Mice

Analysis of leukocyte–endothelium interactions was performed using rhodamine-labeled leukocytes using intravital microscopy. The YNT group presented an increased number of leukocytes rolling at 15 dpi (15.87 ± 4.81 cells/min), considering the NINT control group (1.93 ± 1.57 cells/min, *p* < 0.001). The same pattern of YNT was observed in the Y + LOV group (14.71 ± 5.41 cells/min). When counting adherent leukocytes, both YNT and Y + LOV groups (5.99 ± 2.27 and 6.03 ± 2.54 cells/min, respectively) (Figure 3C,D) had higher numbers than the NINT control group (0.53 ± 0.65 cells/min—*p* < 0.001) at 15 dpi (Figure 3A). These results indicate that therapy with lovastatin did not prevent leukocyte–endothelial interactions. The NI + LOV group presented no variation in comparison to NINT (Figure 3B).

### 3.4. Influence of Lovastatin on the Cerebral Blood Flow Reduction Induced by Acute T. cruzi Infection in Mice

Next, we used laser speckle contrast imaging (LSCI) for a non-invasive real-time evaluation of cerebral vascular perfusion, to assess brain blood flow. This resource measures brain blood flow in arbitrary perfusion units (APUs). Non-infected animals presented brain blood flow values of 218.08 ± 34.58 APUs (Figure 4). During infection, a decrease in brain blood flow was observed in the YNT group, and was not changed by the lovastatin treatment (162.75 ± 30.65, *p* < 0.01 and 151.55 ± 24.68 APUs, respectively) at 15 dpi (Figure 4C,D). The brain blood flow of the NI + LOV animals did not change when compared to the NINT control group (Figure 4B).

### 3.5. Influence of Lovastatin Treatment on the Increase in F4/80+ and CD3+ Cells in the Brain Caused by Acute T. cruzi Infection in Mice

To evaluate the presence of inflammatory infiltrate in mice’s brains, we analyzed the leukocyte subpopulation CD3+ and macrophage F4/80+. We verified that the number of CD3+ cells in the brain increased in the YNT group (3.02 ± 1.38 cells/mm^2^) at 15 dpi (Figure 5C) when compared to the NINT control group (0 cells/mm^2^, *p* < 0.0001) (Figure 5A). Lovastatin did not present a significant change in the increase in CD3+ cells (1.77 ± 0.96 cells/mm^2^) (Figure 5D).

Otherwise, analyzing F4/80+ cells, we observed that *T. cruzi* infection induces an increase in the number of these cells in the YNT group at 15 dpi (2.34 ± 0.89 cells/mm^2^) (Figure 6C) when compared to the NINT control group (0.13 ± 0.14 cells/mm^2^, *p* < 0.0001) (Figure 6A). Lovastatin treatment prevented this increase caused by the acute infection (1.23 ± 0.4 cells/mm^2^, *p* < 0.01) (Figure 6D). No alterations in both CD3+ and F4/80+ cells were observed between NI + LOV and NINT groups.

### 3.6. Impact of Lovastatin on the Increased Inflammatory Mediators ICAM-1 and VCAM-1 in the Brain during Acute T. cruzi Infection in Mice

Once we previously observed increased leukocyte recruitment and an increased number of inflammatory cells in the brain of infected animals [17,18], we investigated if the expression of intracellular adhesion molecule type 1 (ICAM-1) and vascular cell adhesion molecule type 1 (VCAM-1) was affected. These are important markers of endothelial activation, involved in the transport of leukocytes through the vascular wall [13].

ICAM-1 levels were higher in the YNT group (1.67 ± 0.26 V.R. *p* < 0.01) when compared to NINT (1 ± 0.28 F.C.) (Figure 7A). Lovastatin treatment was able to prevent this increase (1.17 ± 0.31 F.C., *p* < 0.05). While evaluating VCAM-1 levels, no significant changes were observed (NINT 1 ± 0.38.; YNT 1.35 ± 0.47; Y + LOV 1.08 ± 0.23 F.C.) (Figure 7B). The NI + LOV animals presented no change in VCAM-1 or ICAM-1 level when compared to the NINT control group.

### 3.7. Influence of Lovastatin Treatment on MCP-1 Levels in the Brain during Acute T. cruzi Infection in Mice

Since we showed increased macrophage recruitment [18], we aimed to verify the levels of monocyte chemoattractant protein-1 (MCP-1), an important protein involved in the migration and infiltration of monocytes/macrophages at the site of inflammation [22].

Acute *T. cruzi* infection did not induce any alteration in MCP-1 levels in the brain of mice (NINT 1 ± 0.25; YNT 0.92 ± 0.23; Y + LOV 0.74 ± 0.30 F.C.) (Figure 7C). The NI + LOV animals had no change in MCP-1 levels when compared to the NINT control group.

### 3.8. Effect of Lovastatin on eNOS Levels in the Brain during Acute T. cruzi Infection in Mice

Our previous studies showed endothelial dysfunction [17]. Here, we investigate the levels of the enzyme endothelial nitric oxide synthase (eNOS), responsible for the production of nitric oxide (NO) in blood vessels and is involved in the regulation of vascular function [28].

We observed high levels of eNOS in the brain of infected animals (2.48 ± 0.54 F.C.) when compared to non-infected animals (1 ± 0.49 F.C. *p* < 0.01). However, lovastatin treatment does not affect these protein levels during infection (2.04 ± 0.84 F.C.) (Figure 8). Non-infected animals treated with LOV had no change in eNOS levels compared to the NINT control group.

## 4. Discussion

In the present study, we investigated the role of lovastatin treatment in microvasculopathy and inflammation in the brain of Swiss Webster mice during acute infection with the Y strain of *T. cruzi*. We assessed the number of spontaneously perfused capillaries, leukocyte–endothelium interaction (rolling and adhesion), cerebral blood flow, presence of inflammatory cells, the expression of inflammatory mediators, and the enzyme eNOS responsible for the control of endothelial function.

The infection of male Swiss Webster mice with 10^4^ trypomastigote forms of the Y strain resulted in the peak of parasitemia at 8 dpi and a low survival rate, as we described here, and are in accordance with previous studies using the same model [17,18]. Treatment protocols were adapted from previous studies on the effect of statins during the acute phase of experimental CD [25,26], starting 1 day after infection and maintained for 14 days. Treatment with LOV did not change the parasitemia of infected animals when compared to untreated infected animals, and did not affect the high mortality observed in this acute experimental model of CD, corroborating the data of Urbina and collaborators [25].

On the other hand, treatment with simvastatin at a dose of 20 mg/kg/day in a model of acute CD using C57BL/6 mice infected with the Colombian strain, demonstrated a reduction in parasitemia and increased survival rate [26]. Since the Colombian strain is TcI and the Y strain is TcII, some differences between them are expected. Studies indicate that TcII could be more infective [27], and even in regions where the presence of TcI and TcII strains are observed in the vectors, chronic patients only present TcII strains [29]. Perhaps this effect could be restricted to simvastatin and specific to the Colombian strain.

We verified the decrease in the number of spontaneously perfused capillaries and the increased rolling and adhesion of leukocytes in the brain of infected animals at 15 dpi. Treatment with LOV had no effect on these parameters, in contrast to what was observed in an experimental model of cerebral malaria using mice infected with *P. berghei* [30]. Reis and collaborators observed that treatment with LOV, at the same dose used here, was able to reverse brain microvasculopathy caused by infection [30]. However, treatment began after the animals started to show clinical signs (between 3 and 6 days after infection) which would correspond to 9–10 dpi in our study.

Although LOV has beneficial effects on vascular function [23], using our animal model of CD acute infection, treatment was not able to prevent the decreased brain blood flow induced by infection. Meningoencephalitis is a severe condition that affects mainly children and immunosuppressed patients during acute CD [12]. In a CD patient’s brain, inflammatory nodules in resolution and inflammatory cells can be observed with a multifocal distribution [12,13,31]. Through immunohistochemistry, we evaluated the presence of CD3+ cells (lymphocytes) and F4-80+ (macrophages) in the brains of infected mice. We verified that *T. cruzi* infection leads to an increase in both cell populations in the brain tissue. In immunocompetent mice C3H/He, infection with *T. cruzi* (Colombian strain) resulted in the presence of inflammatory infiltrates composed mainly of CD8+ T lymphocytes in the brain tissue. Also, macrophages and CD4+ T cells can be found in smaller amounts, during the acute phase of the infection [13].

Using a different acute model of CD, Silva and collaborators showed that treatment with simvastatin decreased inflammatory infiltrate in the heart of mice during *T. cruzi* infection using H&E [26]. Also, another study using dogs infected with the Y strain of *T. cruzi* observed that simvastatin treatment led to reduced inflammation in the heart tissue [28]. However, there is no study indicating the effect of statins on different leukocyte populations on CD acute experimental infection. Interestingly, our data evaluating the effect of lovastatin on brain tissue inflammation demonstrated that treatment prevented the increase in F4/80+ cells. Even with the tendency of a decrease in CD3+ cells observed in the Y + LOV group, this change was not significant in the statistical test, possibly due to the reduced number of animals per group and a high variation between the results for each sample.

Inflammatory mediators play a fundamental role in the migration of leukocytes to the central nervous system (CNS). Inflammatory cells adhere to the endothelium in order to complete the diapedesis process; therefore, the action of integrins such as LFA-4 and VLA-4, which respectively bind to ICAM-1 and VCAM-1, is fundamental [32].

Here, we verified the effect of LOV on these mediators and observed that treatment prevented the increase in ICAM-1 levels, but VCAM-1 and MCP-1 levels were not affected by either infection or treatment. The effect of statins on inflammatory mediators had already been described in CD. Studies demonstrated that simvastatin decreased the expression of ICAM-1 and VCAM-1 in the cardiac tissue in both acute [28] and chronic [33] experimental models of CD.

Furthermore, a study described an increased expression of VCAM-1 in blood vessels surrounded by VLA-4+ inflammatory cells, leading to meningoencephalitis during the acute phase of experimental CD. In this study, functional experiments demonstrated that the VLA-4/VCAM-1 pathway is involved in the entry of inflammatory cells into the CNS [32].

In addition, acute *T. cruzi* infection promoted increased levels of cytokines such as TNF-α, IL-10, IFN-γ, and CCL2/MCP-1 in the brain of rats [34]. Also, lovastatin was able to reduce the MCP-1 renal levels in an ischemia and reperfusion model using rats [35].

Finally, *T. cruzi* infection led to increased eNOS levels. In CD, studies showed that different NOS isoforms can be observed and are related to the early phase of the infection in experimental acute Chagas cardiomyopathy [36]. However, there is evidence of reduced levels of eNOS in the aorta of mice during the acute *T. cruzi* infection [37].

The literature indicates a protective role of LOV in endothelial shear stress through the release of endothelial nitric oxide synthase (eNOS), once NO is a key component for normal vascular tone [38]. However, in our study, eNOS levels were already increased by infection, and treatment had no effect.

Since the heart is the main organ committed by *T. cruzi* infection, some differences can be observed between organs while analyzing inflammatory parameters. The study of how the brain is affected by the parasite is relatively new, and only a few studies explore this [11,12]. Also, using the Swiss Webster + Y strain pair, our group is the first to describe such alterations [17,18]. There is much more to be explored in future studies, and we hope to answer more gaps with regard to the effect of the acute *T. cruzi* infection on the brain.

Taken together, our data indicate an anti-inflammatory activity of LOV during acute *T. cruzi* infection, observed by prevention of the increase in F4-80+ and ICAM-1 levels in the brains of treated infected animals. However, more analysis is needed to elucidate the role of LOV treatment in CD acute infection.

## 5. Conclusions

In this paper, we show that using the pair Swiss Webster mice + Y strain of *T. cruzi*, lovastatin was able to prevent the increase in macrophage number and ICAM-1 levels in the brain tissue caused by the acute infection. In conclusion, our data suggested that lovastatin has some protective effect on the brain of *T. cruzi*-infected mice, and more studies are needed to elucidate this action.

## Figures and Tables

**Figure 1 biology-13-00301-f001:**
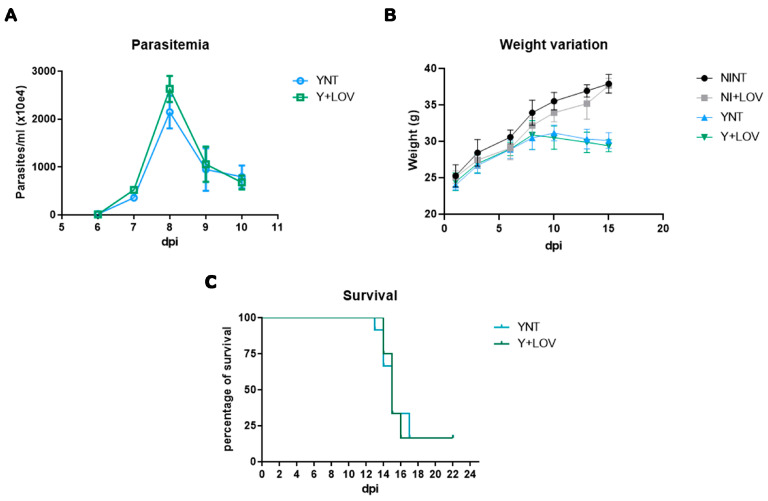
Experimental model (Swiss Webster/Y strain) characterization of (**A**) parasitemia, (**B**) weight, and (**C**) survival. The parasitemia peak was verified at 8 dpi, which was not affected by lovastatin treatment (**A**). Non-significant changes were observed in the body of the YNT group, which was not affected by lovastatin treatment (**B**). Treatment had no effect on survival rate (**C**). NI, non-infected mice; Y, infected; NT, non-treated; LOV, treated with lovastatin (20 mg/kg/day); dpi: days post infection. *n* = 4 animals per experiment in 3 independent experiments.

**Figure 2 biology-13-00301-f002:**
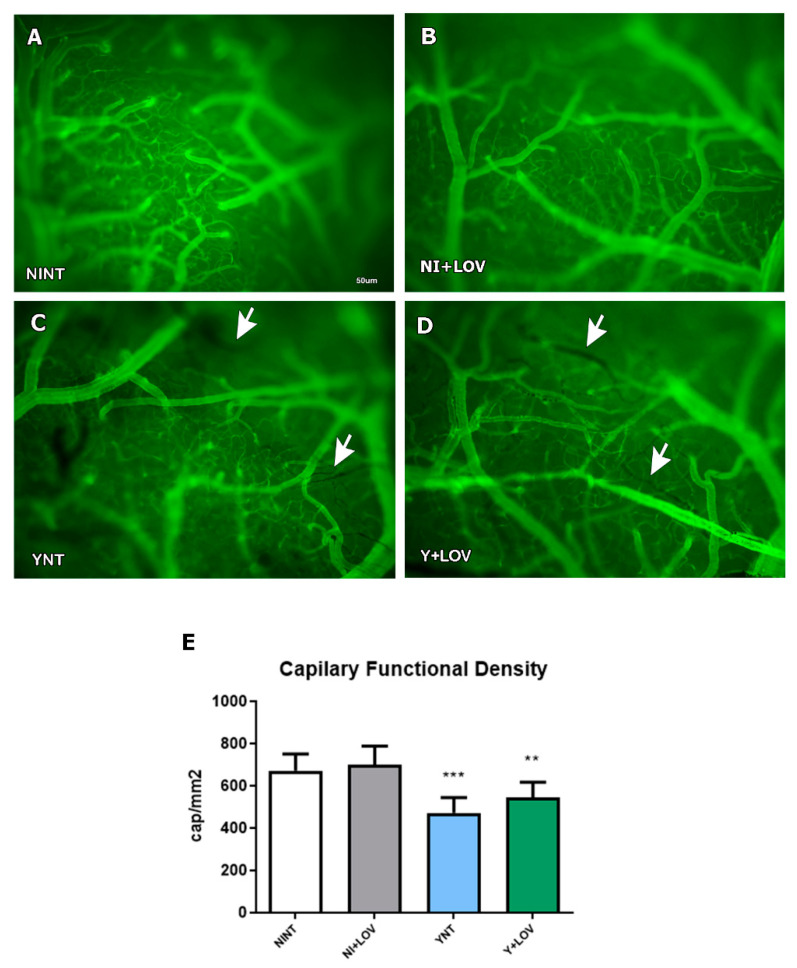
Analysis of brain capillary perfusion using intravital microscopy at 15 dpi. In YNT group (**C**), a decreased number of capillaries were verified, in comparison with NINT group (**A**). Lovastatin had no effect on this parameter in infected (**D**) and non-infected (**B**) animals. Graphic indicates the number of perfused capillaries observed (**E**). Arrows indicate non-perfused NI (non-infected mice); Y, infected; NT, non-treated; LOV, treated with lovastatin (20 mg/kg/day); dpi: days post infection. Asterisk compared to NINT; ** *p* < 0.01, *** *p* < 0.001. *n* = 4 animals per experiment in 3 independent experiments.

**Figure 3 biology-13-00301-f003:**
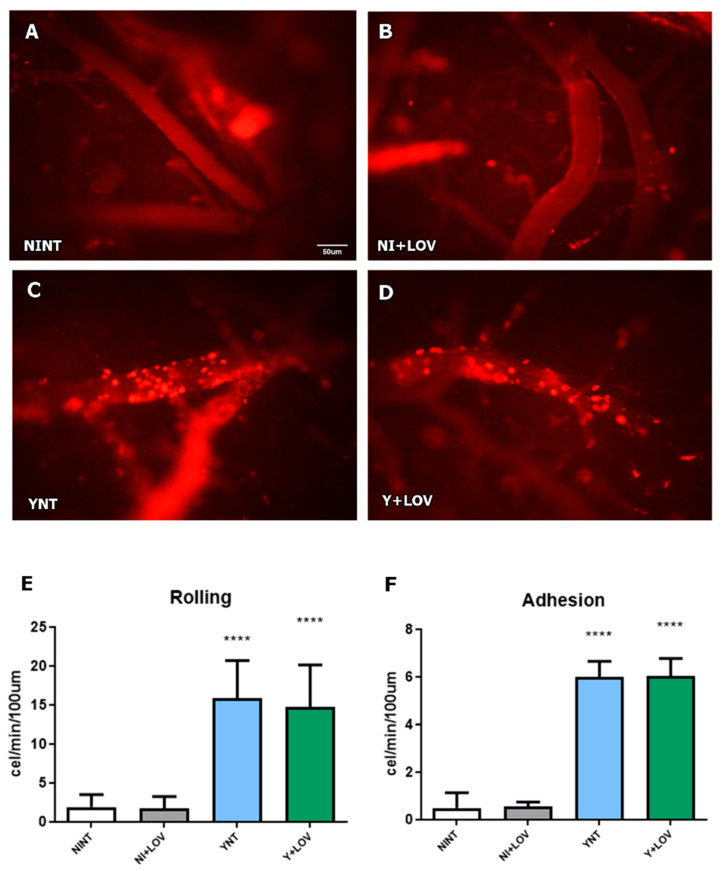
Analysis of leukocyte rolling and adhesion in cerebral venules of mice by intravital microscopy at 15 dpi. We verified a high number of leukocytes rolling and adhering to the venules in the YNT group (**C**) in comparison to the NINT group (**A**). Lovastatin did not alter the rolling and adhesion pattern caused by *T. cruzi* infection (**D**) and had no effect on this in non-infected animals (**B**). Graphic indicates the number of leukocytes rolling (**E**) and adherent (**F**) to venules. NI, non-infected; Y, infected; NT, non-treated; LOV, treated with lovastatin (20 mg/kg/day); dpi, days post infection. Asterisk compared to NINT; **** *p* < 0.0001. *n* = 4 animals per experiment in 3 independent experiments.

**Figure 4 biology-13-00301-f004:**
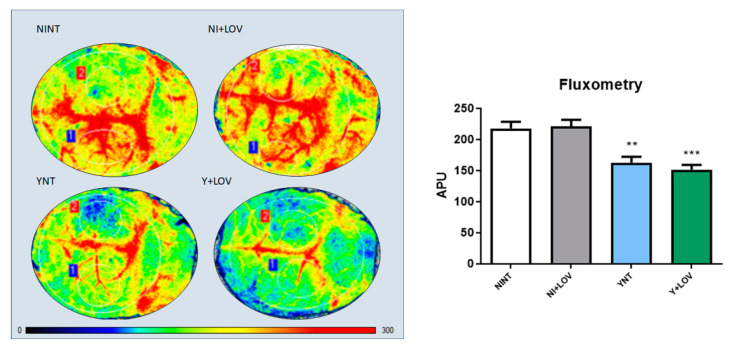
Flowmetry to evaluate cerebral blood flow of infected mice at 15 dpi. A decrease in cerebral blood flow was observed in the YNT group in comparison to NINT group. Treatment with lovastatin had no effect on this in Y + LOV and NI + LOV animals. Graphic shows cerebral blood flow analysis through arbitrary perfusion units (APUs). The numbers 1 and 2 correspond to the area from the craniotomy, which was used for this analysis. NI, non-infected; Y, infected; NT, non-treated; LOV, treated with lovastatin (20 mg/kg/day); dpi, days post infection. Asterisk compared to NINT; ** *p* < 0.01, *** *p* < 0.001 *n* = 3/4 animals per experiment in 3 independent experiments.

**Figure 5 biology-13-00301-f005:**
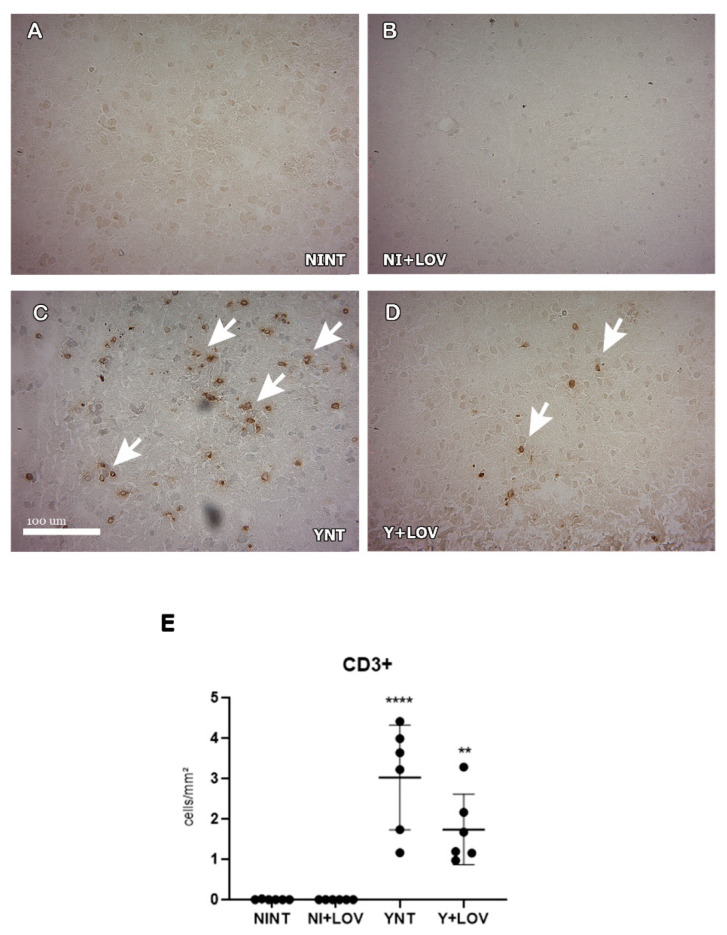
Evaluation of the presence of CD3+ cells in mice’s brain using immunohistochemistry at 15 dpi. (**A**–**D**) Representative images of CD3+ staining for each group. CD3+ cells were not found in NINT group (**A**). YNT group (**C**) presented a higher number of CD3+ cells in comparison to NINT. Treatment with lovastatin in infected (**D**) and non-infected (**B**) animals presented no difference when compared with the respective non-treated group. Graphic indicates the amount of CD3+ cells per mm^2^ (**E**). Arrows show positive cells. NINT, non-infected non-treated; NI + LOV, non-infected treated with lovastatin (20 mg/kg/day); YNT, infected non-treated; Y + LOV, infected treated with lovastatin (20 mg/kg/day); dpi, days post infection. Asterisk compared to NINT; ** *p* < 0.01, **** *p* < 0.0001. *n* = 3 animals per experiment in 2 independent experiments.

**Figure 6 biology-13-00301-f006:**
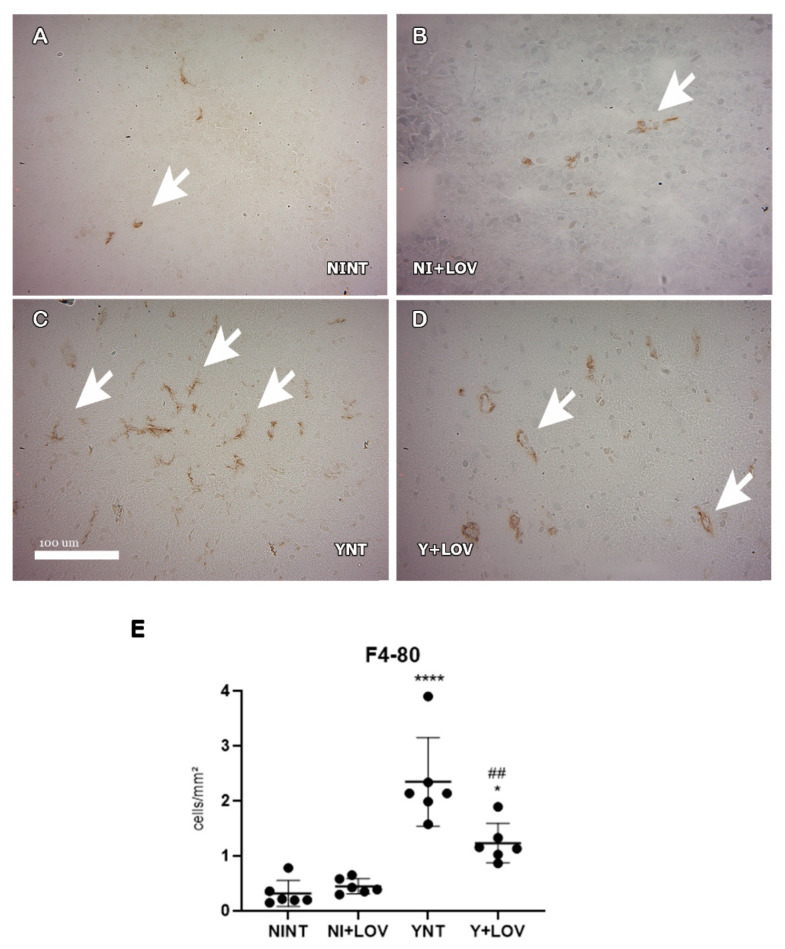
Evaluation of the presence of F4/80+ cells in mice’s brain using immunohistochemistry at 15 dpi. (**A**–**D**) Representative images of F4/80+ staining for each group. A low number of F4/80+ cells were found in NINT group (**A**). YNT group (**C**) had an increase in the number of F4/80+ cells in comparison to NINT. Lovastatin treatment was able to prevent this increase in infected animals (**D**). Treatment had no effect on non-infected animals (**B**). Graphic indicates the number of F4/80+ cells per mm^2^ (**E**). Arrows indicate positive cells. NINT, non-infected non-treated; NI + LOV, non-infected treated with lovastatin (20 mg/kg/day); YNT, infected non-treated; Y + LOV, infected treated with lovastatin (20 mg/kg/day); dpi, days post infection. Asterisk compared to NINT; Sharp compared to YNT; * *p* < 0.05, ## *p* < 0.01, **** *p* < 0.0001. *n* = 3 animals per experiment in 2 independent experiments.

**Figure 7 biology-13-00301-f007:**
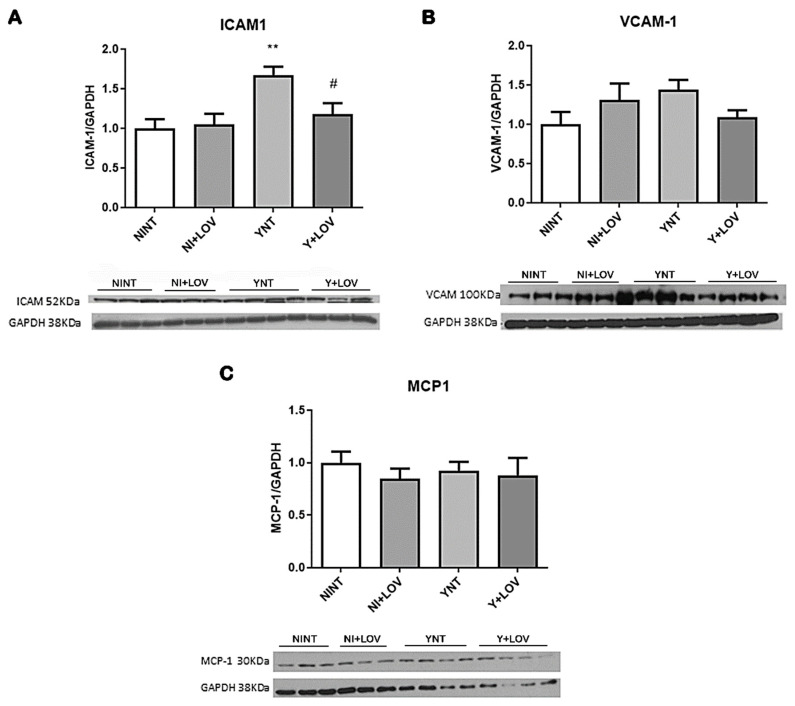
Analysis of inflammatory mediators by Western blot in mice’s brain at 15 dpi. Infection led to an increase in ICAM-1 levels in YNT group; lovastatin treatment prevented this increase (**A**). Neither infection nor lovastatin treatment altered VCAM-1 levels (**B**) and MCP-1 levels (**C**). NI, non-infected; Y, infected; NT, non-treated; LOV, treated with lovastatin (20 mg/kg/day); dpi, days post infection. Asterisk compared to NINT; Sharp compared to YNT; # *p* < 0.05, ** *p* < 0.01. *n* = 3/4 animals per experiment in 2 independent experiments.

**Figure 8 biology-13-00301-f008:**
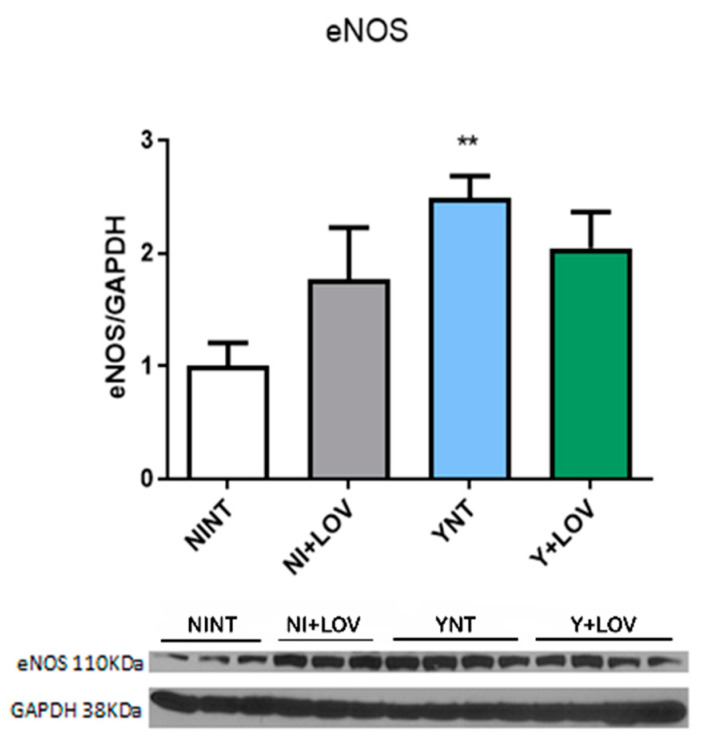
Analysis of the impact of *T. cruzi* infection on endothelial nitric oxide synthase expression by Western blot. An increase was observed in YNT when compared to the NINT group. Lovastatin treatment had no effect on this. NI, non-infected; Y, infected; NT, non-treated; LOV, treated with lovastatin (20 mg/kg/day); dpi, days post infection. Asterisk compared to NINT; ** *p* < 0.01. *n* = 3/4 animals per experiment in 2 independent experiments.

**Table 1 biology-13-00301-t001:** Antibodies used for Western blot analysis (Appendix A).

Antibodies Used for Western Blot	Manufacturer	Catalog Number	Dilution	Protein Concentration
Primary antibody: Anti-ICAM-1	Abcam	ab124760	1:1000	20 μg
Primary antibody: Anti-VCAM1	Abcam	ab115135	1:1000	20 μg
Primary antibody: Anti-MCP-1	Abcam	ab702	1:1000	20 μg
Primary antibody: Anti-eNOS	Abcam	ab5589	1:1000	100 μg
Primary antibody: Anti-GAPDH	Fitzgerald	10R-G109a	1:40,000	-
Secondary antibody: Anti-rabbit with peroxidase	Thermo Scientific	31,430	1:10,000	-
Anticorpo secundário de cabra	Thermo Scientific	31,460	1:10,000	-

## Data Availability

The data obtained and analyzed during the current study are available from the corresponding author upon reasonable request.

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
