# Peer review of "Unveiling Lovastatin’s Anti-Inflammatory Potential in Mouse’s Brain during Acute Trypanosoma cruzi Infection"

_biology, 2024, doi:10.3390/biology13050301_

Round 1

Reviewer 1 Report

Comments and Suggestions for Authors

This study aims to assess the effect of early lovastatin treatment in an acute model of Chagas disease characterised by cerebral microvasculopathy. The authors assess the effect of lovastatin treatment in disease severity, brain capillary density, leukocyte cytoadhesion to endothelial cells, blood flow, brain F4/80+ and CD3+ cell populations, expression of endothelial cell ICAM1 and VCAM1, levels of MCP1 and eNOS. Despite many negative results, experiments are interesting and well performed, so I consider the article relevant for publication. However, I believe some improvements would be ideal, including one further experiment and a few text clarifications.

1)     Starting 1 day post-infection is very early. Why was this chosen? It appears to lack clinical relevance as it’s not early enough for a diagnosis and also disagrees with the protocol used in P. berghei, where they actually showed an effect of lovastatin treatment (discussed in lines 393 to 398).

2)     Is capillary functional similar across brain regions? IVM targets surface

3)     Line 174 – typo “definin the” should be “defining”

4)     Could the authors please adjust scientific notation to the standard nomenclature throughout the paper? For example: 2145 x 104 parasites/ml should be 2.145 x 107 parasites/ml (line 215).

5)     In Figure 5, there’s an apparent decrease in CD3+ cells in the Y+LOV group. However, the authors state that there is no difference. Is this a matter of significance? I also notice that the experiments groups in this part are composed of only 3 mice, over 2 independent experiments.  Statistics with such low numbers of mice are always tricky. I suggest to add a sentence acknowledging this. Also, in Figure 6E, a decrease of apparently the same order of magnitude is shown and determined significant, so I assume it’s a matter of variation. Therefore, a strong suggestion is to show the actual data points with or without the bars rather than just the average values in all graphs.

6)     Lines 313 to 315 are confusing. As it’s written, it comes across that the prevention of ICAM1 expression increase caused by lovastatin treatment leads to the prevention of VCAM1 increase. But VCAM1 is not significantly increase even in the absence of lovastatin.  Why would the authors assume ICAM1 and VCAM1 expression levels are dependent on each other? If this is not intended, then I suggest removing the “therefore” or rephrasing.

7)     Line 382-385: Can the authors elaborate on the works cited here and how they differ from the current work? As it’s written it seems that it’s exactly the same thing.

8)     Line 390-391: what’s different between lovastatin and simvastatin? Do they have the same mode of action? Can the authors speculate on what could explain the differences in outcome? Parasite strain specificities also? What is the Colombian strain? I think this is important for the non-CD specialist to understand the line of thought.

9)     The authors show no significant effect on CD3+ levels upon lovastatin treatment, but it is known that the majority of brain infiltrates are composed of CD8+ T cells. Can the authors check if there is a change in CD8 and CD4+ T cells in the brain of mice upon lovastatin treatment? If it affects CD4+, but not CD8, it would explain the lack of change in disease severity, whilst still be in line with the anti-inflammatory effect of the drug. It would also help to explain why, even with a reduction in ICAM1 expression, similar levels of lymphocyte infiltration are observed.

10)  From the discussion, it becomes apparent that much of what was observed for cardiac pathology is not being reproduced in the brain (at least in this study), namely increase in VCAM1, effect of eNOS on ECs. I would like to see a couple of sentences stating how the differences observed may suggest organ-specific pathology and why the brain might be different (BBB regulation?)

11)  Do the mice definitely die of brain disease? I assume this is in the authors previous paper, but it would be good to clarify it right at the beginning of the results section and again in the discussion.

12)  Line 432: typo “DC” rather than “CD”.

Reviewer 2 Report

Comments and Suggestions for Authors

Round 2

Reviewer 1 Report

Comments and Suggestions for Authors

I would like to thank the authors for a detailed and considered response letter.

A single comment left of a detected typo:

"Since Colombian strain is TcI 413 and Y strain is TcII, some differences between THEN are expected" It should be "THEM". 

Author Response

Thank you for the contributions. 

The typo was corrected in the text.

Reviewer 2 Report

Comments and Suggestions for Authors

Despite the author's efforts to improve the quality of the manuscript, important problems remain with the quality of the presentation, i.e. Figures 5 and 6 show the same images and the same graph. The manuscript is not acceptable in its present form.

Comments on the Quality of English Language

English quality is acceptable

Author Response

Thank you for the comments.

Figure 5 with CD3+ analysis was replaced in the manuscript.

We appologize for the mistake.